# Neutrophil Extracellular Traps and Neutrophil-Derived Extracellular Vesicles: Common Players in Neutrophil Effector Functions

**DOI:** 10.3390/diagnostics12071715

**Published:** 2022-07-14

**Authors:** Heiko Pfister

**Affiliations:** Munich Biomarker Research Center, Institute of Laboratory Medicine, German Heart Center Munich, Technical University Munich, D-80636 Munich, Germany; pfister@dhm.mhn.de

**Keywords:** extracellular vesicles, neutrophil extracellular traps, neutrophil granulocytes, cancer, autoimmune disease, thrombosis

## Abstract

Neutrophil granulocytes are a central component of the innate immune system. In recent years, they have gained considerable attention due to newly discovered biological effector functions and their involvement in various pathological conditions. They have been shown to trigger mechanisms that can either promote or inhibit the development of autoimmunity, thrombosis, and cancer. One mechanism for their modulatory effect is the release of extracellular vesicles (EVs), that trigger appropriate signaling pathways in immune cells and other target cells. In addition, activated neutrophils can release bactericidal DNA fibers decorated with proteins from neutrophil granules (neutrophil extracellular traps, NETs). While NETs are very effective in limiting pathogens, they can also cause severe damage if released in excess or cleared inefficiently. Since NETs and EVs share a variety of neutrophil molecules and initially act in the same microenvironment, differential biochemical and functional analysis is particularly challenging. This review focuses on the biochemical and functional parallels and the extent to which the overlapping spectrum of effector molecules has an impact on biological and pathological effects.

## 1. Neutrophil Extracellular Traps

Upon activation, neutrophil granulocytes can release decondensed chromatin, which forms a scaffold of extracellular fibers decorated with granule proteins. Originally, this structure was thought to function as a trap, killing invading microorganisms, and was given the designation neutrophil extracellular trap (NET) [1]. Although numerous studies have demonstrated that comparable structures may also be produced by eosinophils, mast cells, and macrophages, neutrophil-derived extracellular traps have been the most intensively studied [2,3,4].

NET formation is triggered by contact with a variety of pro-inflammatory stimuli. These include molecules collectively referred to as “pathogen-related molecular patterns” (PAMPs) derived from bacterial, fungal, viral, or parasitic pathogens, e.g., lipopolysaccharides (LPS) or N-formylated chemotactic oligopeptides (e.g., N-formyl-methionyl-leucyl-phenylalanine (fMLP)). Intracellular proteins and extracellular matrix proteins released by stressed or dying cells represent another group of pro-inflammatory signaling molecules (summarized as “damage-related molecular patterns” (DAMPs)). Neutrophils carry specific pattern recognition receptors (PRRs) on their cell membrane that bind PAMPs or DAMPs and trigger a signal transduction cascade, leading to the formation of NETs. PRRs involved in neutrophil NETosis include the families of Toll-like receptors (TLRs), nucleotide-binding oligomerization domain (NOD)-like receptors, and C-type lectin receptors, with each receptor triggered by a distinct set of stimuli [5].

In the canonical pathway of NET generation, contact of PRRs with compounds derived from pathogenic microorganisms, immune complexes, cytokines, or mitogens, including phorbol-12-myristate-13-acetate (PMA), initiates mitogen-activated protein kinases (MAPKs)-dependent signaling pathways that induce the generation of reactive oxygen species (ROS). The subsequently activated peptidyl-arginine-deiminase 4 (PAD-4) and the granule enzymes neutrophil elastase (NE) and myeloperoxidase (MPO) induce chromatin de-condensation by modifying histone proteins [6]. Citrullination by PAD enzymes and enzymatic degradation of histones destabilizes the chromatin structure. As a result, chromatin undergoes a process of entropic swelling that leads to rupture of the nuclear and cell membrane and release of decondensed chromatin [7]. Since the physical swelling of chromatin results in the lysis of a netting cell (a cell releasing NETs) within 2 to 4 h after stimulation, this type of NETosis has been termed “suicidal” NETosis. It represents a distinct cell death pathway, different from classical pathways such as apoptosis, and is characterized by NADPH-oxidase (NOX)-mediated ROS generation [8].

Another non-lytic mode of NET generation is characterized by a rapid cellular response to an external stimulus that occurs within minutes. Nuclear DNA (nDNA)-containing vesicles are formed by budding of a decondensing nucleus and are then transported to the cell membrane [9]. Remarkably, in this process, termed “vital” NETosis, neutrophils become anuclear cytoplasts that are still motile and can sense chemotactic gradients [10]. A third mode of NET generation, which is ROS-dependent and “non-suicidal”, has been described by Yousefi et al. [11]. When granulocyte/macrophage-colony stimulating factor (GM-CSF)-primed neutrophils are stimulated with LPS or complement factor 5a, they release NETs exclusively derived from mitochondrial DNA (mtDNA, Figure 1).

At this point, it should be emphasized that there is still an ongoing debate in the scientific community concerning key aspects of NETs arising from conflicting data. Currently, there is no consensus on the origin of NET-DNA and the exact sequence of events leading to NET formation and cell death [12]. However, an important role of NETs in the areas described in the following sections is supported by a large body of evidence published by various research groups.

The comparability of different studies on NET biology and pathophysiology is often limited because widely accepted standardized procedures for neutrophil stimulation, NET isolation, and characterization have not been established. Recently, a detailed proteomic analysis of NETs revealed a stimulus-dependent pattern of protein composition and post-transcriptional modifications [13]. NETs may even be identified by a differential proteomic signature that appears to be associated with particular diseases [14]. Neutrophil heterogeneity and inter-individual variations, including varying predisposition for NETotic activity, pose a challenge for the identification of specific, functionally relevant differences between NETs that have been generated in different pathways and are composed of DNA from diverse subcellular origins (nuclear, mitochondrial, or both) [15,16]. Pieterse et al. could demonstrate that NETs generated by a NOX-dependent mechanism have a reduced capacity to activate endothelial cells compared to NOX-independent NETs [17]. However, it is unclear whether N-terminally truncated core histones found in NOX-dependent NETs have a significant effect in vivo. Considering the differences of mtDNA and nDNA in size and structure, it is reasonable to assume that NETs derived from either source might differ in their structural composition and thus also in their function [18]. However, further studies are required to clarify to what extent the structural differences are functionally involved in the development of diseases and how they can be integrated into diagnostic procedures.

## 2. Extracellular Vesicles

Classically, remote intercellular communication is achieved by the release of soluble molecules into the extracellular space and subsequent (receptor-mediated) uptake by recipient cells. In contrast to this simple mode of delivering effector molecules, extracellular vesicles (EVs) serve as carriers for multiple cytosolic proteins, nucleic acids, and lipids protected by a lipid bilayer, as well as transmembrane and membrane-associated proteins that contribute to cellular targeting [19]. The release of EVs represents a fundamental mechanism of cell-to-cell communication in eukaryotic and prokaryotic organisms, including Archaea [20]. Originally considered membrane debris with no biological function, EVs are today recognized as crucial mediators of intercellular communication in health and disease [21].

The term extracellular vesicle describes a group of membrane-enveloped vesicles of varying size and genesis that are shed by a variety of cells, including endothelial cells, blood cells, and cancer cells, often as a result of activation or cell death. Three major classes of EVs are distinguished [22]: (1) exosomes (30–150 nm in size) released by the fusion of multivesicular endosomes with the cell membrane, (2) microvesicles (100–1000 nm) formed by budding of the cell membrane, and (3) apoptotic bodies (50–5000 nm) released by dying cells [23] (Figure 1). EVs have recently gained increasing attention in the scientific literature because they appear to play an important role in inflammation, thrombosis, and cancer. Since EVs are found in circulating blood and in various types of extracellular fluids, they are ideal candidates for new diagnostic and therapeutic approaches [24].

Communication between immune cells and other cells involved in immune reactions is a prerequisite to orchestrate an appropriate response to infection and the resolution of inflammation and its sequelae. Neutrophil granulocytes, representing the first line of defense against infectious agents, are able to phagocytose, directly kill, and digest microbial pathogens through the action of various enzymes and proteins stored in their granules [25]. In addition, they release EVs either spontaneously or in response to various priming or activating stimuli. These stimuli include bacteria and fungi, endogenous inflammatory mediators (cytokines and chemokines, complement components), and pharmacologic agents (ionophores, phorbolesters) [26]. The stimulation of EV release, though, is not inevitably linked to the activation of neutrophils. While LPS-induced EV release appears to be related to neutrophil degranulation, GM-CSF-induced EV release is not [27]. In fact, many single-receptor stimulants do not enhance EV release from neutrophils. Activation of PRRs by fMLP or LPS or engagement of Fc-receptors by immunoglobulins alone does not lead to enhanced EV release, but additional co-stimulation of complement receptors induces a marked increase of EV shedding [28].

Depending on the stimulus and the physiologic state, neutrophils are able to release EVs with even opposing functional properties [29]. By targeting different cell types with specific pro- or anti-inflammatory molecules, neutrophils modulate immune responses of both the innate and adaptive immune system. For example, azurophilic granule enzymes, Annexin A1, and the membrane phospholipid phosphatidylserine (PS) can modulate dendritic cell and T-cell activity [30].

Recently, microparticles with an elongated shape (termed elongated neutrophil-derived structures, ENDS) released by rolling neutrophils in the vessel lumen have been reported [31]. Mass spectrometric analysis revealed an EV-like proteome, although classical EV markers such as tetraspanins CD9, CD63, or CD81 could not be detected. Unlike EVs, ENDS do not appear to contain DNA. Plasma levels of ENDS are greatly increased in septic patients and may be involved in the pathology of sepsis through the release of a pro-inflammatory DAMP protein complex consisting of S100A8-S100A9. However, current data are insufficient to define the role of ENDS in inflammatory processes in vivo.

## 3. Common Players: NETs and Neutrophil-Derived EVs

Neutrophils undergo a complex stepwise activation process during their recruitment to sites of injury or infection, while receiving inflammatory signals from their environment [32]. The idea that EVs reflect the activation state of the cell is supported by the fact that EVs from stimulated neutrophils have been shown to exhibit a wide range of functional diversity and target numerous different cell types [33,34,35]. Unlike NETs, which are released upon cell activation, EVs are produced by both resting and activated neutrophils. EVs from resting or apoptotic neutrophils preferentially have an anti-inflammatory effect, possibly contributing to the resolution of inflammation and preventing autoimmunity [26]. Whether an activated cell releases pro- or anti-inflammatory EVs depends largely on the type and amount of stimulus. However, the degree of activation does not seem to be the sole decisive factor in determining the EV type. Neutrophils infected with *Mycobacterium tuberculosis* produce EVs that exert a much stronger activating effect on macrophages than EVs from neutrophils activated with the highly potent NETosis inducer PMA, as evidenced by the induction of pro-inflammatory cytokines and the expression of costimulatory molecules by monocyte-derived macrophages [36].

A dichotomous mode of immune modulation observed with EVs has also been described for NETs: Despite their well-documented pro-inflammatory activity, NETs tend to form inflammation-resolving aggregates at a high cellular density in neutrophil-rich inflammatory foci [37,38]. NET aggregation protects NET-associated NE from inhibition by its physiologic inhibitor α1-antitrypsin (α1-AT). The preserved proteolytic activity exerts an anti-inflammatory effect by degrading pro-inflammatory cytokines and chemokines.

Since both NETs and EVs are released simultaneously upon activation, the question arises of whether they act as antagonistic, synergistic, or independent mediators. Given their common cellular origin, it is not surprising that both share a broad spectrum of molecules involved in the regulation of immune responses in a narrow or broader sense (Table 1). Nonetheless, due to quantitative differences in effector molecule load, different accessibility to interacting molecules, differing cellular and molecular targeting, and clearing routes, considerably different and even opposing functions of the same molecule derived from either NETs or EVs can be anticipated. However, published data suggest that NETs and EVs do not act completely independently of each other in these aspects either: Wang et al. were able to prove a histone-PS-mediated binding of EVs to NETs from murine neutrophils stimulated with PMA or streptococcal M1 protein [39]. By inhibiting EV formation with caspase and calpain inhibitors, they demonstrated that EVs bound to NETs contribute substantially to neutrophil attraction and prothrombotic activity of NETs [39,40]. It is interesting to note that, at least under static conditions, heterologous EV-NET complexes can also be formed. 4T1 breast cancer cell-derived EVs can bind to NETs and appear to synergistically augment the pro-coagulative and prothrombotic capacity of NETs by delivering tissue factor (TF) [41,42,43,44].

The biological significance of these “EV-NET hybrids” remains elusive. In view of the sparse data available on this subject, it seems expedient to intensify research into their function and clearance pathways. In mice, EVs are cleared from the circulation within a day and then accumulate primarily in the liver [69,70]. The half-life of biotin/luciferase double-labeled EVs in blood has been estimated to be approximately 3 h [71]. In contrast, NETs may persist for at least several days in the vasculature [72]. DNase1 and DNase1-like 3 play a pivotal role in the clearing process by degrading the NET-DNA backbone, followed by phagocytic uptake by macrophages and dendritic cells [73,74]. Consequently, DNase1 and DNase1-like 3 may release NET-associated EVs into the circulation as complexes with DNA fragments, thus affecting their cellular targeting and clearance (Figure 2). It cannot be excluded that the binding of EVs to NETs even influences the NET architecture. Detailed analysis by atomic force microscopy revealed that NET proteins are crucial for the structural integrity of NETs [75]. It can be speculated that EVs influence regular NET structure through proteins exposed on their membrane, comparable to NET-proteins crosslinked by polyamines [76]. The remarkable overlap in the protein equipment of NETs and EVs and the resulting potential functional overlaps have not yet been satisfactorily addressed in the scientific literature. Considering the enormous pathogenic potential of NETs and EVs, it is evident how vitally important it is to study these compounds in a more differentiated way at the biochemical and functional levels.

## 4. “Impact beyond Shelf Life”: NETs and Neutrophil-Derived EVs in Disease

Molecules common to NETs and EVs include enzymes with pleiotropic functions such as NE, a major granule serine protease that acts as a pro- and anti-inflammatory effector molecule and is involved in the process of NETosis itself. NE drives chromatin de-condensation by proteolytic cleavage of nuclear proteins during NETosis and during an equivalent process in macrophages [38,77,78,79,80]. The proteolytic activity of NE mediates basic bactericidal and immune modulatory effects, but also contributes to the pathology of various diseases, including autoimmunity, cancer, cardiovascular, and pulmonary diseases (Table 1 and Table 2) [81,82]. Dysregulated release or impaired clearance of NETs and EVs may play a key role by triggering etiopathological mechanisms [83]. The delivery of NET proteases, including the three major azurophilic serine proteases: NE, Cathepsin G, and Proteinase-3 (PR-3), by EVs or NETs in the context of chronic inflammatory lung disease is well-documented [84,85]. Excess NE contributes to disease pathogenesis by activating airway remodeling, interfering with elements of the innate immune system, and triggering pro-inflammatory signaling [80]. In the context of COVID-19 infection, NE contributes to bronchial epithelial cell disruption by diminishing mucociliary transport. Additionally, NE and MPO degrade heparan sulfate, an important structural component of lung parenchyma. In this way, NET/EV-associated proteases directly contribute to acute lung injury in SARS-CoV-2 disease [86].

### 4.1. Autoimmune Disease

The pathogenic potential of proteases shared by NETs and EVs is not limited to a role as effector molecules based on catalytic and direct biochemical functions. Like DNA and histones, they can serve as autoantigens in various autoimmune diseases. Antibodies against MPO and PR-3 are a diagnostic hallmark of anti-neutrophil cytoplasmic antibody (ANCA)-associated vasculitides (AAVs) [128]. Netting neutrophils surrounding small vessels are thought to contribute to vessel inflammation in AAV in two ways: (1) NETs as a direct actor by damaging endothelial cells and (2) NETs as autoantigens with high immunogenicity that triggers ANCA production. ANCAs in turn can stimulate the release of neutrophil EVs and induce NETosis of primed neutrophils [117,129]. In addition to excessive NET formation, reduced DNAse1 activity and elevated NET levels in the serum of AAV patients indicate that incomplete NET clearance may be involved in disease pathogenesis [129,130]. Neutrophil-derived PR-3- and MPO-expressing EVs may promote small-vessel vasculitis by triggering an inflammatory cascade and directly damaging endothelial cells through transfer of microRNAs (miRNAs) [117,118]. It is unclear to what extent neutrophil-derived EVs may augment inflammation in AAVs by presenting NET autoantigens as in other autoimmune disorders: Patients with systemic lupus erythematosus (SLE) develop autoantibodies to various NET constituents, which together can form immune complexes, causing nephritis. Delayed removal of NETs due to impaired DNAse1-mediated digestion of the DNA backbone likely promotes the presentation of self-antigens, which initiates the process of SLE [121]. EVs are an additional source of autoantigen in SLE [131]. They have been shown to form non-classical EV-containing immune complexes that are related to disease activity [132]. However, a prominent SLE-associated EV population bearing a neutrophil protein signature has not yet been reported [133,134]. Citrullinated vimentin and citrullinated fibrinogen are major autoantigens in rheumatoid arthritis (RA) [135]. Again, NETosis appears to play a pivotal role in the loss of self-tolerance [136]. Equivalent to SLE, pro-inflammatory complexes of autoantigen-presenting EVs and autoantibodies may be involved in pathogenesis [137]. Potentially immunogenic (or tolerogenic) EVs from various cellular origins carrying the respective autoantigen are found in a variety of further autoimmune diseases [138,139,140]. Pauci-immune glomerulonephritis typically associated with AAVs argues against disease-relevant autoantigen-specific EV-antibody complexes in AAV. In contrast, they might be supported by an electron microscopic analysis of renal biopsies from AAV patients, which revealed weak immune complex deposits in the majority of cases examined [141]. It is unclear whether reported cases of ANCA-negative pauci-immune glomerulonephritis in RA and SLE are associated with EV-antibody complexes or whether they are etiologically different forms of SLE [142,143,144].

### 4.2. Cancer

Distinct functional plasticity of neutrophils, which is evident in the immunological context, is also reflected in their multifaceted effects on carcinogenesis and the spread of cancer [145]. In the tumor microenvironment, transforming growth factor-β (TGF-β), interferon-β (IFN-β), and granulocyte-colony-stimulating factor (G-CSF) polarize neutrophils towards a pro- or anti-tumorigenic phenotype. Polarized neutrophils modulate tumor growth, progression, and metastasis through a variety of mechanisms. The basis for neutrophil diversity is already laid during neutrophil development in the bone marrow, where neutrophil progenitor cells arise from a presumably heterogeneous pool of granulocyte-monocyte progenitor cells [113]. Granulopoesis and release from the bone marrow can be affected by tumors and tumor-associated cells that release cytokines and chemokines. Consequently, neutrophils of different developmental stages can be found in the circulation [146]. It has been reported that immature circulating neutrophils have a higher global bioenergetic capacity, which facilitates sustained NETosis [147]. NETs are known to play a vital role in fostering pro-tumorigenic neutrophil activity. They can promote tumor growth by either directly inducing cancer proliferation or by shielding tumors from cytotoxic immune cells [148,149,150]. In addition to the inflammatory triggers discussed in the previous sections, NETosis can also be induced by factors released from tumor cells. G-CSF released from 4T1 cells has been found to induce NET formation, resulting in cancer cell invasion in vitro [151]. Furthermore, Cathepsin C released from breast cancer cells can induce NETosis by activating a zymogenic form of NE on the neutrophil cell membrane. Cathepsin C-induced NETs, in turn, have been shown to degrade the metastasis-suppressing protein thrombospondin-1 (TSP-1), promoting metastasis [152]. Neutrophil-triggered degradation of TSP-1 was also shown to be a pro-metastatic mechanism in a mouse model in which tobacco smoke-induced neutrophil NETs awakened dormant breast cancer cells. In this experimental setting, NET-associated NE and matrix metalloproteinase-9 sequentially cleaved laminin, and the processed laminin induced integrin-mediated proliferation of dormant cancer cells [153]. Interestingly, the DNA-scaffold provided by NETs appears to support the proteolytic laminin cleavage by binding to laminin. NET-DNA can therefore be considered a “carrier” for proteolytic neutrophil enzymes. However, DNA itself has been shown to possess pro-metastatic potential as well: NET-DNA exerts chemotactic activity towards cancer cells and is able to capture them from the circulation. Stimulation of the DNA-binding receptor CCDC25 on cancer cells by NET-DNA activates the integrin-linked kinase β-parvin pathway, resulting in increased cell motility [154,155].

While NETs are primarily attributed with pro-tumorigenic activity, EVs originating from anti-tumorigenic neutrophils (termed N1 neutrophils) and pro-tumorigenic neutrophils (termed N2 neutrophils) are reflecting the current cell status [26,146,156]. The anti-tumor activity of EVs is mediated by the release of pro-inflammatory cytokines (IL-1β, IL-2, IL-4), molecules that modulate cell motility and adhesion, such as integrins, granule enzymes, and others [35,63,126,157]. Granule proteins shared by NETs and neutrophil-derived EVs are also an important component of the neutrophil’s pro-carcinogenic arsenal. NE can induce cancer proliferation, migration, and invasion. Although not mandatory, the presence of NE detected in an endosomal compartment in lung tumor cells is consistent with the assumption that NE could be transmitted by EVs [158,159]. An imbalance of the inhibitor and protease apparently favors cancer cell development, growth, and migration [160,161]. NET-DNA might be a good candidate in this context as a participant in the process of dysregulation of serine protease activity: DNA can reversibly inhibit serine protease activity, protecting it from irreversible inhibition by serpins [38,78,162]. Ultimately, irregular release or impaired clearance of NETs and EVs may have an influence on the fine-tuned protease–anti-protease balance.

Another common major granule enzyme, MPO, was discovered in the circulation of ANCA-associated vasculitis patients as a complex with DNA, presumably derived from NETs [116]. Like NE, MPO also participates in the process of NETosis by facilitating chromatin de-condensation [163]. In this way, MPO is likely to exert an indirect pro-tumorigenic effect. However, MPO itself functions as a modulator of cancer development in multiple steps as well [164]. The role of MPO and NETs in autoimmunity and their involvement in cancer development leads to the hypothesis that there may be a mechanistic link between ANCA and cancer: NETosis-inducing ANCA could initiate a self-perpetuating or reinforcing cycle of NET formation, which in turn triggers autoantibody production and furthermore promotes tumorigenesis [165].

### 4.3. Thrombosis

Remarkably, patients with ANCA-associated autoimmune disease and tumor patients share an increased risk of thrombotic events [115,166]. AAV increases the risk of arterial thrombotic events (ATE) and venous thromboembolism (VTE, including deep vein thrombosis and pulmonary embolism) by at least 2–3-fold [166,167]. Cancer increases the risk of VTE 4- to 7-fold and of ATE approximately 2-fold in the short term [168,169,170]. Although ATE and VTE are classified into two etiologically separate groups, common risk factors for both are well-documented [171,172]. TF has been identified as an important mediator of ANCA-associated thrombosis, which is released packaged in EVs and in NET-bound form by ANCA-activated neutrophils [166]. Vascular injury caused by ANCA-activated neutrophils may lead to additional exposure of TF from adventitial cells that surround blood vessels. By forming a complex with FVII/FVIIa, TF can finally aberrantly trigger blood coagulation, leading to an increased risk of thrombosis [172]. While NET formation appears to be associated with disease activity and thrombosis, TF^+^ EVs increase the risk of VTE regardless of disease activity [173,174]. Autoantibodies not only contribute to excessive NETosis but may also interfere with appropriate NET elimination. Patients with antiphospholipid syndrome (APS), a disorder associated with elevated titers of antiphospholipid and NET autoantibodies, are at increased risk of arterial, venous, and microvascular thrombosis [175]. Antiphospholipid antibodies can activate neutrophils and induce the release of NETs, accelerating thrombosis [176,177]. Anti-NET antibodies may further increase the risk of thrombotic events in APS by binding to NETs, thereby interfering with their proper degradation and clearance [178,179]. In addition to anti-NET antibodies, DNase inhibition may potentially interfere with NET clearance in autoimmune diseases, as suggested by Hakkim et al. for SLE [121].

Cancer cells can predispose patients to thrombosis in at least two ways: (1) Tumor cells directly secrete prothrombotic factors that can initiate blood coagulation by creating a scaffold for blood clotting factors on EVs (through PS), by activating the extrinsic pathway of coagulation (by TF), and by modulating platelet activity (by podoplanin, Adenosine diphosphate (ADP), and thrombin). In addition, they are able to inhibit fibrinolysis by plasminogen activation inhibitor-1 (PAI-1) [115]. (2) Tumor cells can stimulate neutrophils to release NETs [43,180].

NETs can also be directly triggered by specific viral pathogens such as SARS-CoV-2 [181]. NET-induced thrombosis appears to be a key feature of COVID-19 and has been discussed as a major cause of mortality [86].

The biologic function of the prothrombotic effect of NETs can be interpreted in terms of their role as an element of intravascular innate immunity. In the concept of immunothrombosis, the formation of microvascular thrombi helps fight infection and prevent pathogen dissemination and tissue invasion [98]. Neutrophils, and NETs in particular, play a central role in immunothrombosis, involving structural NET components but also proteins found both on NETs and in neutrophil-derived vesicles. NETs can trigger the extrinsic pathway through NET-bound TF and the intrinsic pathway through activation of FXII by the negative surface charge of the DNA backbone [182]. Both pathways finally trigger the common pathway, which results in the formation of thrombin and fibrin deposits. In addition, platelets are bound and activated by histones mediated by von Willebrand factor (vWF). MPO and serine proteases, including NE, have been shown to indirectly promote blood clotting by inhibiting anticoagulants such as tissue factor pathway inhibitor (TFPI) and thrombomodulin [183].

Injury- and infection-related inflammatory processes that result in thrombus formation are also driven by EVs secreted by neutrophils. Like NETs, they can directly activate the extrinsic and intrinsic pathways of blood coagulation: TF and polyphosphates expressed by EVs mediate the activation of FVII and FXII, respectively [184]. PS, an anionic phospholipid shared by NETs and neutrophil-derived EVs, provides a scaffold for the assembly of coagulation complexes on cell membranes. Electrostatic interactions of PS with positively charged residues of clotting proteins facilitate their binding to the lipid bilayer. In addition, PS^+^ EVs are able to induce platelet activation and aggregation [185]. The putative relevance of neutrophil-derived PS^+^ EVs in cardiovascular disease has been demonstrated in non-valvular atrial fibrillation [186]. Thrombosis can additionally be induced by endothelial cell damage caused by the cytotoxic effect of neutrophil EVs carrying MPO [184]. The deleterious effect is probably exerted by enzymatic catalysis of hypochlorous acid (HOCl) generation on the surface of EVs [187]. Likewise, the enzymatic activity of MPO has been demonstrated to contribute to the cytotoxic effects of NETs [188].

## 5. NET- and Neutrophil EV-Based Diagnostics

The important role of NETs and EVs in inflammation and the pathophysiology of autoimmunity, cancer, and thrombosis suggests their use in the development of new diagnostic methods (Table 3). In recent years, procedural approaches to diagnostics based on circulating neutrophil EVs have significantly improved. Initial achievements in the identification of prognostically and diagnostically potentially useful EVs have been published for cardiovascular and pulmonary diseases, among others [189,190,191,192]. However, neutrophil EV-based diagnostics is still in its fledgling stage. Knowledge of molecular EV signatures, including conditional ones that correlate with a biological and pathophysiological context, is sparse. However, it is desirable to exploit the information provided by EVs on the status and environmental cues to which the releasing neutrophils have been exposed. In addition, technical problems such as the limited availability of antibodies and staining techniques with sufficient sensitivity and specificity for newly discovered EV markers compatible with multiplex technologies hamper the development of clinically applicable diagnostics. The first steps towards establishing such methods for detection and characterization have already been taken, e.g., by identifying suitable marker combinations using flow cytometry [193].

Markers of NETosis, on the other hand, may be detected in various, already better-established ways. The concept of liquid biopsy is based on current molecular biology methods such as droplet-based digital PCR and next-generation sequencing technologies. It is used to characterize cell-free DNA (cfDNA) released into the blood and other body fluids by various cells during apoptosis, necrosis, EV-shedding, and as a result of NETosis [208]. Due to the different cellular origins and different cellular processes, it is not feasible to use the release kinetics or levels of cfDNA as NET-related markers. Rather, NET-specific modifications such as citrullinated histones or NET-associated proteins such as MPO can be used as additional markers. Unfortunately, currently, no commercial kit reliably passes the technical difficulties of a clinically validated detection method of DNA-NET-protein complexes [209]. Recently published data even suggest that the widely performed measurement of MPO-DNA complexes by ELISA for the differential determination of NETs is itself questionable, especially in human plasma [210]. The technical challenges of producing reliable measurement results and the lack of standardized methods allowing cross-study analyses also impede the correlation of NET-related data with pathological parameters. Complicating matters further is the conditionally opposing function of NETs, which can act as both pro-inflammatory and anti-inflammatory mediators. Indeed, there are conflicting data, for example, on whether NET levels or NET-degrading activity are correlated with disease activity in autoimmune diseases [130,174,211]. A similar situation is present in cancer diagnostics: although there is some proof of the diagnostic and prognostic relevance of circulating NETs, there is still a lack of solid evidence supported by an adequate amount of clinical data [212].

A combination of NET markers that may be tailored to the experimental or clinical objective could probably improve the quality of study results. Citrullinated histones appear to be a good option as they are directly involved in NETosis. Citrullinated histone H3 in combination with DNA and other NETosis-related markers has been shown to be a suitable analyte for prognosis in cancer, sepsis, or cardiovascular conditions [213,214]. In this context, the formulation and continuous refinement of a guideline based on consensus, comparable to the MISEV guidelines for EVs, would be a valuable aid for the further development of NET-based diagnostics [215].

## 6. Conclusions

The high concordance of protein equipment of NETs and EVs can be explained by their common cellular origin. Accordingly, a good scientific study design and future EV- and NET-based diagnostics should be characterized by a careful selection of NET and EV markers, especially if biological effects of NETs and EVs are to be distinguished. For example, tetraspannins characteristic of EVs (CD9, CD63, CD81) may be included as negative NET markers. To date, there is little data on the extent to which previous contact of EVs with NETs affects their protein equipment and which can be identified by standard immunological detection methods. It is possible that EVs, especially under pathological conditions, carry NET-DNA fragments to which NET-DNA-associated protein is bound. The principal ability of DNA to bind onto the EV surface has been demonstrated with cell lines from T-cell leukemia and pancreatic cancer [216]. As shown by Leal et al., even heterologous EV-NET complexes can form, at least under static conditions [44]. This is both a challenge and an opportunity for future research. The technical methods to distinguish DNA packaged during EV generation and passively released DNA (NETs, apoptotic bodies) are not yet sufficiently established [217]. The challenge is compounded by the large stimulus- and biological context-dependent variability of NETs and EVs. However, suitable techniques could turn (heterologous) EV-DNA complexes into a proper complement to NET- and EV-based diagnostics, especially in the fields of autoimmunity and cancer. Considering the outstanding role of phagocytes in modulating cancer development and autoimmunity, remnants of the contact between neutrophil EVs/NETs and cancer cells as well as other immune cells could help identify pathologic mechanisms and differentiate stages of disease progression, and substantially support therapeutic measures.

## Figures and Tables

**Figure 1 diagnostics-12-01715-f001:**
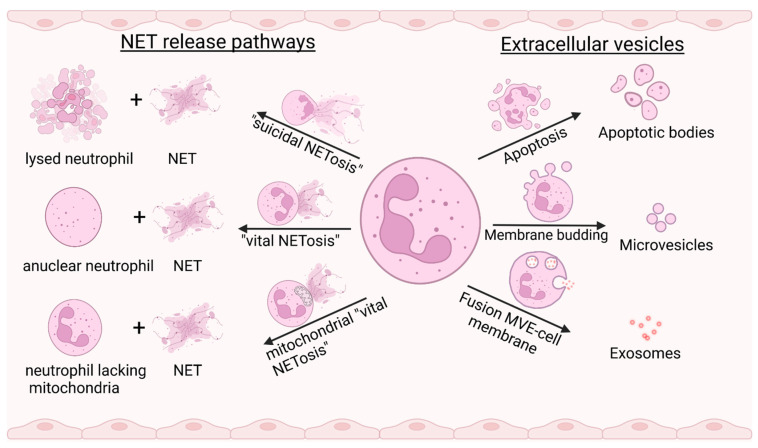
Mechanisms of NET formation (**left**) and EV classes released by neutrophils (**right**). In “suicidal NETosis”, chromatin is released from the lysing neutrophil after chromatin de-condensation and subsequent rupture of the nuclear and cell membrane. “Vital NETosis” is characterized by the fusion of vesicles containing nuclear DNA with the plasma membrane, finally resulting in an anuclear cytoplast. In a second form of “vital NETosis”, neutrophils release mitochondrial DNA by an unknown NOX-dependent mechanism. Extracellular vesicles comprise a group of heterogeneous membranous vesicles of varying size and morphology. Apoptotic bodies represent subcellular fragments after the disassembly of a dying cell. The smaller microvesicles bud from the cell membrane and contain cytoplasmic material. The smallest EVs are referred to as exosomes and are released from the lumen of multivesicular endosomes (MVE), fusing with the cell membrane. Created with BioRender.com.

**Figure 2 diagnostics-12-01715-f002:**
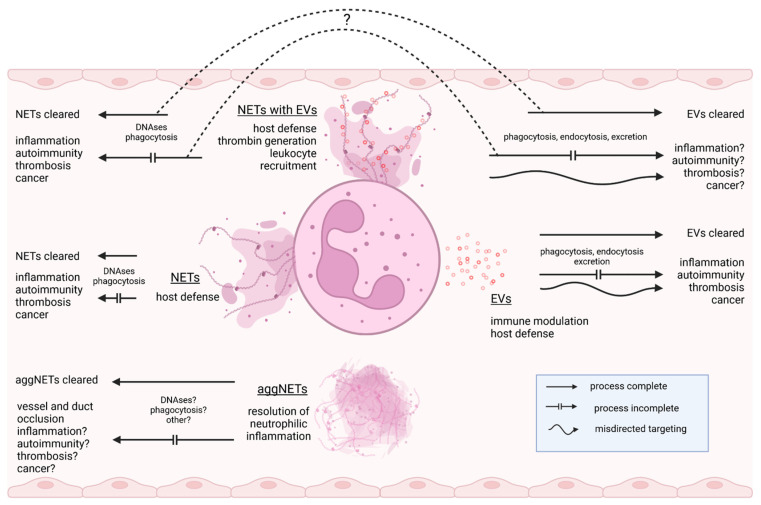
Dysregulated expression or impaired clearance of EVs (by phago-/endocytosis, excretion) and NETs (by DNase-digestion, phagocytosis) may trigger various pathologies. Binding of EVs to NETs may hypothetically lead to a reciprocal influence on EV and NET clearance (dashed line). Created with BioRender.com.

**Table 1 diagnostics-12-01715-t001:** Major substances shared by NETs and EVs/ENDs upon stimulation with various priming or activating agents.

Neutrophil Stimulus Used in Study		Substance Detected with Both NETs and EVs/ENDs	References
EVs/ENDs	NETs		
		Granule proteins/peptides	
*A. fumigatus* conidia	PMA	Azurocidin	[14,45,46,47]
*A. fumigatus* conidia	A23187	Cathelicidin antimicrobial peptide	[14,47]
fMLP	MSU, PMA, IL-8, LPS	Cathepsin G	[8,33,45,46,48]
*A. fumigatus* conidia	A23187	Cysteine-rich secretory protein 3	[14,47]
*A. fumigatus* conidia	TNF-α	Defensins	[47,49]
fMLP, Ionomycin	PMA, IL-8, LPS	Neutrophil Elastase	[1,45,46,50,51]
fMLP, Ionomycin	MSU, PMA, TNF-α, IL-8, LPS	Lactoferrin	[1,33,45,46,48,49,51]
*A. fumigatus* conidia	A23187	Lipocalin	[14,47]
*A. fumigatus* conidia	PMA	Lysozyme C	[46,47]
fMLP, Ionomycin	A23187	Matrix metallopeptidases	[14,50,51]
fMLP, PMA, Ionomycin	MSU, PMA, TNF-α, IL-8, LPS	Myeloperoxidase	[1,8,33,45,46,48,49,51,52]
fMLP	PMA	Proteinase-3	[45,50]
		DAMPs	
Not cell type or stimulus-specific	PMA, IL-8, LPS	DNA	[1,53]
fMLP	MSU, PMA, TNF-α	Histones	[14,33,45,46,48,49]
PMA	PMA	HMGB-1	[40,54]
fMLP	PMA, IL-8, amyloid fibrils, *Leishmania* promastigotes	microRNA	[55,56]
fMLP	PMA, TNF-α,	S100 family proteins	[31,33,45,46,49]
	PMA		
		Cytoskeleton proteins	
fMLP	MSU, TNF-α	Myosin-9	[33,46,48,49]
fMLP	MSU, PMA	Actins	[33,46,48]
fMLP	A23187, MSU, PMA	α-Enolase	[14,31,48]
fMLP, pneumolysin	MSU, PMA	Annexins	[14,33,48,57,58]
fMLP	Rheumatoid Factor	Catalase	[33,59]
fMLP	PMA, TNF-α plus ANCA	Complement components	[33,60,61]
fMLP, LPS	A23187	Cytokines/Chemokines	[14,35,62,63]
fMLP	PMA	Gelsolin	[33,46]
fMLP	Plasma from stroke patients	Phosphatidylserine	[64,65]
Autoimmune vasculitis	Autoimmune vasculitis; deep vein thrombosis; viral infection	Tissue Factor	[66,67,68]

**Table 2 diagnostics-12-01715-t002:** Overlapping and opposing effects of neutrophil-derived extracellular vesicles and NETs in health and disease. NETs and extracellular vesicles trigger a large variety of secondary effector mechanisms that are not entirely included in this table.

Biological Context	Neutrophil EVs	NETs	References
General	May act either as a pro-inflammatory or anti-inflammatory mediator depending on target cells and activation context	May act either as a pro-inflammatory or anti-inflammatory mediator depending on activation context	[87,88]
Complement	Activate complement	Activate complement	[60,89]
Erythrocytes	Bind erythrocytes in the presence of complement	Bind erythrocytes	[89,90]
Monocytes/Macrophages	May induce a pro- or anti-inflammatory response in monocytes/macrophages depending on stimulus	May induce a pro- or anti-inflammatory response in monocytes/macrophages	[26,72,91,92]
Neutrophils	May induce a pro- or anti-inflammatory response in neutrophils depending on stimulus	Pro-inflammatory, and anti-inflammatory in aggregated form	[26,93,94,95,96]
Blood platelets	Activate blood platelets via αMβ2-mediated binding	Activate blood platelets by histones	[97,98]
Endothelial cells	May induce a pro- or anti-inflammatory response in endothelial cells and may promote or reduce para-endothelial permeability depending on stimulus	Activate endothelial cells by Interleukin-1α and Cathepsin G and promote endothelial permeability	[26,99,100]
T-cells	May induce a pro- or anti-inflammatory response in T-cells	May induce a pro- or anti-inflammatory response in T-cells	[35,101,102]
Infection	Antibacterial by: -bacteria aggregation on surface -granule proteins	Antibacterial by: -bacteria entrapment -granule proteins -antimicrobial peptides -histones, DNA	[33,50,103,104]
	No direct evidence for antiviral activity	Antiviral by: -virus entrapment -granule proteins -antimicrobial peptides -histones, DNA	[105]
	Antifungal by granule proteins	Antifungal by calprotectin	[47,106]
	No direct evidence for antiparasitic activity	Antiparasitic by: -entrapment -killing	[107]
Non-autoimmune cardiovascular disease	Promote thrombosis by exposing tissue factor, platelet activating factor, and possibly phosphatidylserine	Promote thrombosis by exposing von Willebrand factor, histones, tissue factor, and phosphatidylserine	[98,103,108,109,110]
	Promote atherosclerosis by delivering microRNA (miR-155)	Promote atherosclerosis by macrophage activation possibly via granule enzymes	[56,111]
Cancer	Anti-tumorigenic by inducing apoptosis of cancer cells or pro-tumorigenic	Pro-tumorigenic, influencing growth, progression, and spreading of cancer by various mechanisms	[112,113,114]
	No direct evidence for cancer-associated pro- or anti-thrombotic effect	May promote cancer-associated thrombosis	[115]
Autoimmunity			
ANCA-associated vasculitis	Promote thrombosis by exposing tissue factor; contains autoantigen; may trigger vasculitis	Promote thrombosis by exposing tissue factor; contains autoantigen; may trigger vasculitis	[66,116,117,118]
Psoriasis	May trigger inflammation	May trigger autoimmunity and inflammation by bound pro-inflammatory IL-17	[119,120]
Systemic lupus erythematosus (SLE)	No evidence for direct involvement in pathogenesis	Contain autoantigen and may contribute to pathogenesis	[121,122,123]
Rheumatoid arthritis (RA)	Protective effect on cartilage	Contain autoantigen and may contribute to pathogenesis of RA; damage cartilage by NE	[49,124,125]
Pulmonary disease	Contribute to disease pathology	Contribute to disease pathology	[85,126,127]

**Table 3 diagnostics-12-01715-t003:** Examples of the potential prognostic and diagnostic use of neutrophil-derived EVs and NETs.

Disease Setting	Study Material	Analyte: NET or Neutrophil EV (Used Markers)	Method	Significance	Refs.
Infection					
Sepsis	Blood	EV (CD15)	Microbead-based isolation + NTA	Level disease-associated + prognostic potential	[194]
Sepsis	Blood	NET formation ex vivo (DNA)	Stimulation of heterologous neutrophils by patient plasma + immunofluorescence microscopic quantification of released DNA	Level disease-associated + prognostic potential	[195]
COVID-19	Blood	EV (PS *, CD15, CD66b)	FC	Level and TF activity associated with thrombotic risk	[196,197]
COVID-19	Blood	NET (MPO-DNA, citrullinated histone, histone H3, cfDNA, NE)	ELISA	Level disease-associated + prognostic potential	[198,199]
Cardiovascular					
Infective endocarditis	Blood	EV (PS *, CD66b)	FC	Level for differential diagnosis and risk assessment	[189]
Infective endocarditis	Blood	NET (MPO-DNA)	ELISA	Level disease-associated	[200]
Unstable plaque in carotid stenosis	Blood	EV (CD11b, CD66b)	FC	Level related to unstable plaque	[190]
Familial hypercholesterolemia	Blood	EV (PS *, CD11b, CD66b)	FC	Combined with EVs from different origins: level correlates with coronary calcification and atherosclerotic plaque	[201]
Coronary artery disease	Blood	NET (dsDNA, nucleosomes, MPO-DNA)	DNA-dye, ELISA	Level correlates with coronary calcification and atherosclerotic plaque	[202]
Lung					
COPD	BALF	EV (CD11b, CD66b)	FC	Level disease-associated	[192]
COPD	Sputum	NET (MPO-DNA, Elastase-DNA, Histone-elastase)	ELISA	Level disease-associated	[203]
ARDS	BALF	EV (CD11b, CD66b)	FC	Level disease-associated	[204]
ARDS	BALF, blood	NET (MPO-DNA)	ELISA	Level disease-associated	[205]
Cancer					
Non-small cell lung cancer	Blood	EV (PS *, CD66b)	FC	Level associated with disease progression	[206]
Various cancers including lung cancer	Blood	NET (citrullinated histone)	ELISA	Level disease-associated + prognostic potential	[207]

* PS detected by Annexin V-staining.

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
