# Peer review of "Neutrophil Extracellular Traps and Neutrophil-Derived Extracellular Vesicles: Common Players in Neutrophil Effector Functions"

_diagnostics, 2022, doi:10.3390/diagnostics12071715_

Round 1

Reviewer 1 Report

Congratulations to the author for this beautiful and comprehensive review on Neutrophil Extracellular Traps (so-called NETs) and Neutrophil Vesicles (EV). In my opinion, the author has dealt with the topic with scientific rigor, providing an abundant and exhaustive bibliography. I also really appreciated the choice of conducting this review in a "narrative" style by dividing it by topic. More in detail, the author focuses on the various types of mechanisms that lead to the formation of NETs / EVs and then deepens their involvement in the setting of various pathologies. Here are my minor comments:

1. Correct some typos that are present.

2. Add a very small paragraph regarding the involvement of NETs / EVs / DAMPs in the setting of Covid-19.

Here are some very recent references:

Veras FP, Pontelli MC, Silva CM, Toller-Kawahisa JE, de Lima M, Nascimento DC, Schneider AH, Caetité D, Tavares LA, Paiva IM, Rosales R, Colón D, Martins R, Castro IA, Almeida GM, Lopes MIF, Benatti MN, Bonjorno LP, Giannini MC, Luppino-Assad R, Almeida SL, Vilar F, Santana R, Bollela VR, Auxiliadora-Martins M, Borges M, Miranda CH, Pazin-Filho A, da Silva LLP, Cunha LD, Zamboni DS, Dal-Pizzol F, Leiria LO, Siyuan L, Batah S, Fabro A, Mauad T, Dolhnikoff M, Duarte-Neto A, Saldiva P, Cunha TM, Alves-Filho JC, Arruda E, Louzada-Junior P, Oliveira RD, Cunha FQ. SARS-CoV-2-triggered neutrophil extracellular traps mediate COVID-19 pathology. J Exp Med. 2020 Dec 7;217(12):e20201129. doi: 10.1084/jem.20201129. PMID: 32926098; PMCID: PMC7488868.

Niedźwiedzka-Rystwej P, Grywalska E, Hrynkiewicz R, Bębnowska D, Wołącewicz M, Majchrzak A, Parczewski M. Interplay between Neutrophils, NETs and T-Cells in SARS-CoV-2 Infection-A Missing Piece of the Puzzle in the COVID-19 Pathogenesis? Cells. 2021 Jul 19;10(7):1817. doi: 10.3390/cells10071817. PMID: 34359987; PMCID: PMC8304299.

Vorobjeva NV, Chernyak BV. NETosis: Molecular Mechanisms, Role in Physiology and Pathology. Biochemistry (Mosc). 2020 Oct;85(10):1178-1190. doi: 10.1134/S0006297920100065. PMID: 33202203; PMCID: PMC7590568.

Szturmowicz M, Demkow U. Neutrophil Extracellular Traps (NETs) in Severe SARS-CoV-2 Lung Disease. Int J Mol Sci. 2021 Aug 17;22(16):8854. doi: 10.3390/ijms22168854. PMID: 34445556; PMCID: PMC8396177.

Cicco S, Cicco G, Racanelli V, Vacca A. Neutrophil Extracellular Traps (NETs) and Damage-Associated Molecular Patterns (DAMPs): Two Potential Targets for COVID-19 Treatment. Mediators Inflamm. 2020 Jul 16;2020:7527953. doi: 10.1155/2020/7527953. PMID: 32724296; PMCID: PMC7366221.

Cazzato G, Colagrande A, Cimmino A, Cicco G, Scarcella VS, Tarantino P, Lospalluti L, Romita P, Foti C, Demarco A, Sablone S, Candance PMV, Cicco S, Lettini T, Ingravallo G, Resta L. HMGB1-TIM3-HO1: A New Pathway of Inflammation in Skin of SARS-CoV-2 Patients? A Retrospective Pilot Study. Biomolecules. 2021 Aug 16;11(8):1219. doi: 10.3390/biom11081219. PMID: 34439887; PMCID: PMC8392002.

Author Response

I thank you for taking the time to assess the manuscript and greatly appreciate your thorough and thoughtful comments and suggestions. The revision of the manuscript according to your suggestions has improved its quality.

Overview of the changes made:

  • Review and correction of grammatical errors and typos.
  • Insertion of topic-relevant information on Covid-19.
  • Addition of a figure to illustrate the different NET formation pathways as well as the different EV classes.
  • Inclusion of a table on the use of NETs and EVs as diagnostic/prognostic markers for various diseases.
  • Update of the literature list to include the additional references from the revisions 2) and 4).

Attached below are detailed responses to your suggestions. The latter are shown in black and responses in red.

  1. Correct some typos that are present.

The manuscript was carefully checked for typos and corrected accordingly.

  1. Add a very small paragraph regarding the involvement of NETs / EVs / DAMPs in the setting of Covid-19.

Information on Covid-19 has been included. In order to maintain the conceptual structure of the manuscript, the new passages can be found in the appropriate context at various points in the manuscript. A single paragraph would not fit as well into the flow of the text.
Please see page 7, lines 236-240, page 12, lines 375-377, and page 13, lines 14-19 in table 3 for inserted passages.

I would like to thank the referee again for taking the time to review the manuscript.

Reviewer 2 Report

This is a comprehensive review article on the functions of NETs and EVs in physiological and pathological conditions. Each section is concise and does not contain redundant information. The author also includes relevant diagram and tables that summarizes the respective sections. Overall, the manuscript was written in good English and easy to follow and understand although there are still minor typos and grammatical errors. For instance, line 55 ",,suicide"" should be corrected as "suicide". Please check the entire manuscript again minor spelling/typo/grammatical errors. A minor comment that I would recommend is to provide a brief diagram that summarizes the pathways of NET generation as well as different classes of EVs as described in sections 1 and 2. This will help the readers to understand the biology of NETs and EVs.  

Author Response

I thank you for taking the time to assess the manuscript and greatly appreciate your thorough and thoughtful comments and suggestions. The revision of the manuscript according to your suggestions has improved its quality.

Overview of the changes made:

  • Review and correction of grammatical errors and typos.
  • Insertion of topic-relevant information on Covid-19.
  • Addition of a figure to illustrate the different NET formation pathways as well as the different EV classes.
  • Inclusion of a table on the use of NETs and EVs as diagnostic/prognostic markers for various diseases.
  • Update of the literature list to include the additional references from the revisions 2) and 4).

Attached below are detailed responses to your suggestions. The latter are shown in black and responses in red.

  1. Please check the entire manuscript again minor spelling/typo/grammatical errors.

The manuscript was carefully checked for typos and corrected accordingly.

  1. A minor comment that I would recommend is to provide a brief diagram that summarizes the pathways of NET generation as well as different classes of EVs as described in sections 1 and 2. This will help the readers to understand the biology of NETs and EVs.

On page 3, a figure has been included that schematically shows the different NET forming pathways and the different classes of EVs. A caption has been added for explanation.

I would like to thank the referee again for taking the time to review the manuscript.

Reviewer 3 Report

In this review, the author summarizes current knowledge about neutrophil extracellular traps (NETs) and neutrophil extracellular vesicles (EVs), comparing their role and involvement in different diseases (autoimmunity, cancer, thrombosis). In addition, he discussed the possibility to use these structures in the diagnostics.

The strengths of the review are that all NETosis pathways have been analyzed, all extracellular vesicles have been described, and their independent, antagonistic or synergistic action has been shown.

The manuscript is well structured, based on recent literature resources and good interpretation. Information is presented in a logical way, illustrated with one figure and two tables. The conclusions are clear and concrete, and the usage of the NETs and EVs compounds as diagnostic markers is clearly defined.

The conclusions are clear and concrete, and the use of compounds of NETs and EVs as diagnostic markers is clearly defined.

Recommendation:

It will be good to show as a figure or in a table the NETs/EVs markers that are (or could be) used as diagnostic markers of certain diseases.

Author Response

I thank you for taking the time to assess the manuscript and greatly appreciate your thorough and thoughtful comments and suggestions. The revision of the manuscript according to your suggestions has improved its quality.

Overview of the changes made:

  • Review and correction of grammatical errors and typos.
  • Insertion of topic-relevant information on Covid-19.
  • Addition of a figure to illustrate the different NET formation pathways as well as the different EV classes.
  • Inclusion of a table on the use of NETs and EVs as diagnostic/prognostic markers for various diseases.
  • Update of the literature list to include the additional references from the revisions 2) and 4).

Attached below are detailed responses to your suggestions. The latter are shown in black and responses in red.

It will be good to show as a figure or in a table the NETs/EVs markers that are (or could be) used as diagnostic markers of certain diseases.

A table has been included on page 13-14 showing the relevance of NETs and EVs for diagnostic and prognostic purposes in various diseases

I would like to thank the referee again for taking the time to review the manuscript.